# Mercury Chloride Affects Band 3 Protein-Mediated Anionic Transport in Red Blood Cells: Role of Oxidative Stress and Protective Effect of Olive Oil Polyphenols

**DOI:** 10.3390/cells12030424

**Published:** 2023-01-27

**Authors:** Pasquale Perrone, Sara Spinelli, Gianluca Mantegna, Rosaria Notariale, Elisabetta Straface, Daniele Caruso, Giuseppe Falliti, Angela Marino, Caterina Manna, Alessia Remigante, Rossana Morabito

**Affiliations:** 1Department of Precision Medicine, School of Medicine, University of Campania Luigi Vanvitelli, 80138 Naples, Italy; 2Department of Chemical, Biological, Pharmaceutical and Environmental Sciences, University of Messina, 98122 Messina, Italy; 3Biomarkers Unit, Center for Gender-Specific Medicine, Istituto Superiore di Sanità, 00161 Rome, Italy; 4Complex Operational Unit of Clinical Pathology, Papardo Hospital, 98122 Messina, Italy

**Keywords:** mercury chloride, oxidative stress, band 3 protein, anion exchange, human RBCs

## Abstract

Mercury is a toxic heavy metal widely dispersed in the natural environment. Mercury exposure induces an increase in oxidative stress in red blood cells (RBCs) through the production of reactive species and alteration of the endogenous antioxidant defense system. Recently, among various natural antioxidants, the polyphenols from extra-virgin olive oil (EVOO), an important element of the Mediterranean diet, have generated growing interest. Here, we examined the potential protective effects of hydroxytyrosol (HT) and/or homovanillyl alcohol (HVA) on an oxidative stress model represented by human RBCs treated with HgCl_2_ (10 µM, 4 h of incubation). Morphological changes as well as markers of oxidative stress, including thiobarbituric acid reactive substance (TBARS) levels, the oxidation of protein sulfhydryl (-SH) groups, methemoglobin formation (% MetHb), apoptotic cells, a reduced glutathione/oxidized glutathione ratio, Band 3 protein (B3p) content, and anion exchange capability through B3p were analyzed in RBCs treated with HgCl_2_ with or without 10 μM HT and/or HVA pre-treatment for 15 min. Our data show that 10 µM HT and/or HVA pre-incubation impaired both acanthocytes formation, due to 10 µM HgCl_2_, and mercury-induced oxidative stress injury and, moreover, restored the endogenous antioxidant system. Interestingly, HgCl_2_ treatment was associated with a decrease in the rate constant for SO_4_^2−^ uptake through B3p as well as MetHb formation. Both alterations were attenuated by pre-treatment with HT and/or HVA. These findings provide mechanistic insights into benefits deriving from the use of naturally occurring polyphenols against oxidative stress induced by HgCl_2_ on RBCs. Thus, dietary supplementation with polyphenols might be useful in populations exposed to HgCl_2_ poisoning.

## 1. Introduction

Human exposure to heavy metals has increased dramatically in the last five decades due to exponential increase in their use in various industries and products [1,2]. Among heavy metal pollutants of the natural environment, mercury is the most common. There are three forms of mercury: elemental (or metallic) mercury, inorganic mercury and organic mercury [3]. Although mercury is present in low concentration in nature, it poses a health risk to human beings and other organisms due to its persistency, ingestion of contaminated fish (such as swordfish, tuna or shark), bioaccumulation, and toxicity [4,5].

Mercury preferentially accumulates in red blood cells (RBCs), thus inducing morphological changes and an increase in the pro-coagulant activity of these cells [6,7]. Red blood cells are unique cells regarding their structural organization and function. Although their primary function is the transportation of the respiratory gases O_2_ and CO_2_ between lungs and tissues, these circulatory cells are equipped with efficient endogenous anti-oxidative systems that make them mobile free radical scavengers providing antioxidant protection, not only to RBCs themselves but also to other tissues and organs in the body [8]. However, these anucleated cells can be affected, to different extents, by oxidative injuries whenever oxidative stress develops. For example, the incubation of RBCs with varying concentrations of mercury results in the externalization of phosphatidylserine (PS), cell eryptosis, and morphological perturbations in cell shape [9,10,11]. The underlying mechanism of mercury toxicity is correlated with its high affinity for sulfhydryl groups which severely alter enzyme activities as well as membrane structural proteins, such as Band 3 protein (B3p). Band 3 protein, or anion exchanger 1 (AE1), is encoded by the *SLC4A1* gene and, with more than 1 million copies per cell, is the most abundant membrane protein in RBCs. The crystal structure of B3p has been recently obtained, revealing two domains, an N-terminal cytosolic domain that anchors the cytoskeleton at the plasma membrane and interacts with different proteins and a C-terminal membrane domain mediating the anion exchange [12]. Since an imbalance in the physicochemical properties of RBCs can make them dysfunctional and impede efficient tissue oxygenation, the maintenance of their functionality is of the outmost importance [13]. Mercury exposure has been proven to cause hemolysis, thus representing a risk of anemia [14,15,16]. In fact, mercury is able to trigger a complex reaction that leads to cell damage and finally cell death. When mercury enters RBCs, a decrease in endogenous antioxidant activity and an increase in pro-oxidant activity are observed, leading to the peroxidative destruction of RBC membranes as well as an increase in osmotic pressure resulting in hemolysis [9,17,18]. Specifically, anemia is associated with a decrease in glutathione peroxidase (GPx) activity [19]. Glutathione peroxidase has an important function in the process of aerobic glycolysis on the pentose phosphate pathway, which is the main energy source for RBCs. Decreasing GPx activity inhibits the glycolysis process, which results in a reduced RBC half-life. Other studies have also demonstrated that mercury exposure increases the production of reactive oxygen species (ROS) on account of the Fenton reaction. Therefore, the hemolytic effects of mercury suggest that RBCs may be an important target of this metal [20,21].

The life functions of RBCs can be potentially improved by functional foods and natural products with antioxidant properties, thus ameliorating the homeostasis of the whole body. The consumption of extra virgin olive oil (EVOO) has been associated with several beneficial healthy effects, partly due to its polyphenol content, known for its important antioxidant activity [22,23,24]. More than 30 phenolic compounds belonging to several polyphenol classes have been identified in EVOO [25]. At a biological level, these compounds are very important not only for their antioxidant activity but also for their capacity in modelling several cellular signaling pathways which exhibit numerous beneficial effects in addition to those directly due to their free radical scavenging activity [26]. It is well known that the stability of cell membranes can be positively affected by exogenous antioxidants [22]. In this regard, several studies have shown a protective effect in terms of olive oil polyphenols, such as hydroxytyrosol (HT) and its metabolite homovanillyl alcohol (HVA), on RBCs (Figure 1a,b) [27,28,29]. 

Since RBCs deliver oxygen to the entire body, the maintenance of a functional and constant amount of RBCs is of pivotal importance for the health of an individual. 

In particular, we explored the protective capacity of HT and its metabolite HVA in a model of oxidative stress represented by human RBCs treated with a non-hemolytic concentration of HgCl_2_ (10 µM). This cell-based model could represent those human pathologic conditions that are hallmarks of chronic mercury toxicity, including hemolytic anemia, and thus affect RBC integrity. Since B3p function has been previously established as a sensitive tool to assess the impact of oxidative stress on the homeostasis of RBCs [30,31,32,33,34,35,36], the anion exchange capability through B3p and the effects of oxidative stress on various cellular components were evaluated.

## 2. Materials and Methods

### 2.1. Solutions and Chemicals

All chemicals were purchased from Sigma (Milan, Italy). 3,4-Dihydroxyphenethyl alcohol (HT, hydroxytyrosol; CAS number: 10597-60-1), 4-hydroxy-3-methoxyphenethyl alcohol (HVA, homovanillyl alcohol; CAS number: 2380-78-1), and 4,4′-diisothiocyanatostilbene-2,2′-disulfonate (DIDS) stock solutions (100 mM for HT and HVA; 10 mM for DIDS) were prepared in dimethyl sulfoxide (DMSO). N-ethylmaleimide (NEM) stock solution (310 mM) was prepared in ethanol. Mercury chloride (HgCl_2_) was diluted in distilled water from a 100 mM stock solution. Both ethanol and DMSO never exceeded 0.001% *v*/*v* in the experimental solutions and were previously tested on RBCs to exclude hemolytic damage.

### 2.2. Preparation of Red Blood Cells

This study was prospectively reviewed and approved by a duly constitute Ethics Committee (prot.52-22, 20-04-2022). Upon informed consent, whole human blood from healthy volunteers was collected in test tubes containing ethylenediaminetetraacetic acid (EDTA). The plasma concentration of glycated hemoglobin (A1c) was less than 5%. Red blood cells were washed in isotonic solution (composition in mM: NaCl 150, 4-(2-hydroxyethyl)-1-piperazineethanesulfonic acid (HEPES) 5, glucose 5, pH 7.4, osmotic pressure 300 mOsm/kgH_2_O) and centrifuged thrice (Neya 16R, 1200× *g*, 5 min) to remove plasma and buffy coat. Red blood cells were then suspended at specific hematocrits in isotonic solution and prepared for downstream analysis.

### 2.3. Analysis of Cell Shape by Scanning Electron Microscopy (SEM)

Samples, which were left untreated or exposed to HgCl_2_ (4 h at 37 °C) or pre-incubated with 10 µM HT and/or HVA (15 min at 25 °C) and then exposed to 10 µM HgCl_2_ treatment, were collected, plated on poly-l-lysine-coated slides and fixed with 2.5% glutaraldehyde in 0.1 M cacodylate buffer (pH 7.4) at 25 °C for 20 min. Then, samples were post-fixed with 1% OsO_4_ in 0.1 M sodium cacodylate buffer and dehydrated through a graded series of ethanol solutions (from 30% to 100%). Absolute ethanol was gradually substituted by a 1:1 solution of hexamethyldisilazane (HMDS)/absolute ethanol and successively by pure HMDS. Afterwards, HMDS was completely removed, and samples were dried in a desiccator. Dried samples were mounted on stubs, coated with gold (10 nm), and analyzed by a Cambridge 360 scanning electron microscope (Leica Microsystem, Wetzlar, Germany) [37]. The altered shapes of RBCs were evaluated by counting ≥ 500 cells (50 RBCs for each different scanning electron microscopy (SEM) field at a magnification of 3000×) from samples in triplicate.

### 2.4. Detection of Reactive Oxygen Species (ROS)

To evaluate intracellular reactive oxygen intermediates, RBCs left untreated or exposed to 10 µM HgCl_2_-containing solutions with or without pre-incubation with 10 µM HT and/or HVA (15 min at 25 °C) were incubated in Hanks’ balanced salt solution, pH 7.4, containing dihydrorhodamine 123 (DHR 123; Molecular Probes, Milan, Italy) and then analyzed with a FACScan flow cytometer (Becton-Dickinson, Mountain View, CA, USA). At least 20,000 events were acquired. The median fluorescence intensity histogram values were used to provide a semi-quantitative analysis of ROS production [38].

### 2.5. Thiobarbituric-Acid-Reactive Substance (TBARS) Level Measurement

TBARS levels were measured as described by Mendanha and collaborators [39], with minor modifications. TBARSs derive from the reaction between thiobarbituric acid (TBA) and malondialdehyde (MDA), which is the end-product of lipid peroxidation [39,40]. Red blood cells were suspended at 20% hematocrit and incubated with 10 µM HgCl_2_ (4 h at 37 °C) or pre-incubated with 10 µM HT and/or HVA (15 min at 25 °C, with or without HgCl_2_ treatment). Then, samples were centrifuged (Neya 16R, 1200× *g*, 5 min) and suspended in isotonic solution. Red blood cells (1.5 mL) were treated with 10% (*w*/*v*) trichloroacetic acid (TCA) and centrifuged (Neya 16R, 3000× *g*, 10 min). TBA (1% in hot distilled water, 1 mL) was added to the supernatant, and the mixture was incubated at 95 °C for 30 min. Finally, TBARS levels were obtained by subtracting 20% of the absorbance at 453 nm from the absorbance at 532 nm (Onda Spectrophotometer, UV-21). Results are indicated as µM TBARS levels (1.56 × 10^5^ M^−1^ cm^−1^ molar extinction coefficient). 

### 2.6. Total Sulfhydryl Group Content 

The measurement of total sulfhydryl group (-SH) groups was carried out according to the method of Aksenov and Markesbery [41], with minor modifications. In short, RBCs (35% hematocrit), left untreated, exposed to 10 µM HgCl_2_ (4 h at 37 °C), or pre-incubated with 10 µM HT and/or HVA (15 min at 25°C, with or without 10 µM HgCl_2_ treatment) were centrifuged (Neya 16R, 1200× *g*, 5 min) and a sample of 100 µL was hemolyzed in 1 mL of distilled water. A 50 μL aliquot was added to 1 mL of phosphate-buffered saline (PBS, pH 7.4) containing EDTA (1 mM). 5,5′-Dithiobis (2-nitrobenzoic acid) (DTNB, 10 mM, 30 μL) was added to initiate the reaction, and the samples were incubated for 30 min at 25 °C while protected from light. Control samples, without cell lysate or DTNB, were processed concurrently. After incubation, the sample absorbance was measured at 412 nm (Onda spectrophotometer, UV-21) and 3-thio-2-nitro-benzoic acid (TNB) levels were detected after the subtraction of the blank absorbance (samples containing only DTNB). To achieve the full oxidation of -SH groups, an aliquot of RBCs (positive control) was incubated with 2 mM NEM for 1 h at 25 °C [42,43]. Data were normalized to protein content and results reported as μM TNB/mg protein. 

### 2.7. Determination of Methemoglobin (MetHb) Levels 

Methemoglobin levels were determined as reported by Naoum and collaborators [44], with minor modifications. The assay is based on MetHb and (oxy)-hemoglobin (Hb) determination by spectrophotometry at a wavelength of 630 and 540 nm, respectively. After incubation (10 µM HgCl_2_ for 4 h at 37 °C with or without pre-incubation with 10 µM HT and/or HVA for 15 min at 25 °C), samples were centrifuged (2000× *g*, 5 min, 25 °C; Eppendorf), and 25 μL of RBCs at 40% hematocrit were lysed in 1975 μL hypotonic buffer (composition: 2.5 mM NaH_2_PO_4_, pH 7.4; 4 °C). Then, samples were centrifuged (13,000× *g*, 15 min, 4 °C; Eppendorf) to eliminate membranes. The absorbance of the supernatant was measured (BioPhotometer Plus; Eppendorf). Incubation with 4 mM NaNO_2_ (for 1 h at 25 °C), a well-known MetHb-forming agent, was used to obtain complete Hb oxidation [45]. The MetHb percentage (%) was determined as follows: % MetHb = (OD630/OD540) × 100 (OD is optical density).

### 2.8. Detection of Apoptotic Red Blood Cells 

Red blood cells, left untreated or exposed to 10 µM HgCl_2_-containing solutions with or without pre-incubation with 10 µM HT and/or HVA (15 min at 25 °C), were processed to detect eryptosis by using the FITC-conjugated Annexin V eryptosis detection kit (Biovision, CA, USA.) and Trypan blue staining (0.05% Trypan blue for 15 min at room temperature) [46]. Then, RBCs were analyzed with a FACS scan flow cytometer (Becton-Dickinson, Mountain View, CA, USA) equipped with a 488 nm argon laser. 

### 2.9. Preparation of Red Blood Cell Membranes

Red blood cell membranes were prepared as described by other authors [47], with slight modifications. Briefly, packed RBCs (untreated or treated with 10 µM HgCl_2_ for 4 h at 37 °C with or without 10 µM or pre-incubated with 10 µM HT and/or HVA (15 min at 37 °C)) were diluted into 1.5 mL of 2.5 mM NaH_2_PO_4_ (cold hemolysis solution) containing a cocktail of inhibitors (1 mM PMSF, 1 mM NaF, and 1 mM Na3VO4). Samples were repeatedly centrifuged (Eppendorf, 4 °C, 18,000× *g*, 15 min) to take out hemoglobin. The obtained membranes were solubilized by sodium dodecyl sulphate (SDS, 1% *v*/*v*) and kept for 20 min on ice. After centrifugation (Eppendorf, 4 °C, 13,000× *g*, 30 min), the supernatant, containing the solubilized membrane proteins, was stored at −80 °C until use.

#### SDS-PAGE Preparation and Western Blotting Analysis 

After thawing, membranes were solubilized in Laemmli buffer (1:1 volume ratio) [48] and heated for 15 min at 95 °C. The protein samples (5 μg/μL), measured by Bradford assay, were separated by 7.5% SDS-polyacrylamide gel electrophoresis and transferred to a polyvinylidene fluoride membrane by applying a constant voltage (75 V) for 2 h at 4°C. Membranes were blocked for 1 h at room temperature in 5% bovine serum albumin (BSA) diluted in Tris-buffered saline (150 mM NaCl, 15 mM Tris-HCl) containing 0.1% Tween-20 (TBST) and incubated overnight at 4 °C with the primary antibody (monoclonal anti-B3p antibody, B9277, Sigma-Aldrich, Milan, Italy, produced in mouse and diluted 1:5000 in TBST). Successively, membranes were incubated for 1 h with peroxidase-conjugated goat anti-mouse IgG secondary antibodies (A9044, Sigma-Aldrich, Milan, Italy) diluted 1:10,000 in TBST solution at room temperature. To assess the presence of equal amounts of protein, a monoclonal anti-β-actin antibody (A1978, Sigma-Aldrich, Milan, Italy), diluted 1:10,000 in TBST solution and produced in mouse, was incubated with the same membrane, as suggested by Yeung and collaborators [49]. A chemiluminescence detection system (Super Signal West Pico Chemiluminescent Substrate, Pierce Thermo Scientific, Rockford, IL, USA) was used to detect signals, whose images were imported to analysis software (Image Quant TL, v2003). The intensities of the corresponding protein bands were determined by densitometry (Bio-Rad ChemiDocTM XRS+).

### 2.10. SO_4_^2−^ Uptake Measurement 

#### 2.10.1. Control Condition

An SO_4_^2−^ uptake measurement was used to evaluate the anion exchange through B3p, as described elsewhere [50,51,52,53]. Briefly, after washing, RBCs were suspended to 3% hematocrit in 35 mL SO_4_^2−^ medium (composition in mM: Na_2_SO_4_ 118, HEPES 10, glucose 5, pH 7.4, osmotic pressure 300 mOsm/kgH_2_O) and incubated at 25 °C in this medium. After 5, 10, 15, 30, 45, 60, 90, and 120 min, DIDS (10 μM), which is an inhibitor of B3p activity [54,55], was added to 5 mL sample aliquots, which were kept on ice. Subsequently, samples were washed three times in cold isotonic solution and centrifuged (Neya 16R, 4 °C, 1200× *g*, 5 min) to eliminate SO_4_^2−^ from the external medium. Distilled water (1 mL) was added to induce the osmotic lysis of RBCs, and perchloric acid (4% *v*/*v*) was used to precipitate proteins. After centrifugation (Neya 16R, 4 °C, 2500× *g*, 10 min), the supernatant containing SO_4_^2−^ trapped by RBCs was directed to the turbidimetric analysis. Supernatant (500 μL from each sample) was sequentially mixed to 500 μL glycerol diluted (1:1) in distilled water, 1 mL 4 M NaCl, and 500 μL 1.24 M BaCl_2_^•^2H_2_O. Finally, the absorbance of each sample was measured at 425 nm (Onda Spectrophotometer, UV-21). The absorbance was converted to [SO_4_^2−^] L cells × 10^−2^ by means of a standard curve previously obtained by precipitating known SO_4_^2−^ concentrations. The rate constant of SO_4_^2−^ uptake (min^−1^) was derived from the following equation: C_t_ = C_∞_ (1 − e^−rt^) + C_0_, where C_t_, C_∞_, and C_0_ indicate the intracellular SO_4_^2−^ concentrations measured at times t, ∞, and 0, respectively, with e representing the Neper number (2.7182818), r indicating the rate constant accounting for the process velocity, and t being the specific time at which the SO_4_^2−^ concentration was measured. The rate constant is the inverse of the time needed to reach ~63% of the total SO_4_^2−^ intracellular concentration [50], and the [SO_4_^2−^] L cells × 10^−2^ reported in figures represents the SO_4_^2−^ micromolar concentration internalized by 5 mL RBCs suspended at 3% hematocrit.

#### 2.10.2. Experimental Conditions 

Samples (3% hematocrit), which were left untreated or exposed to 10 µM HgCl_2_ (4 h at 37 °C) or pre-incubated with 10 µM HT and/or HVA (15 min at 25 °C, with or without HgCl_2_ treatment), were centrifuged (Neya 16R, 4 °C, 1200× *g*, 5 min) to replace the supernatant with SO_4_^2−^ medium. The rate constant of SO_4_^2−^ uptake was then determined as described for the control condition.

### 2.11. Measurement of Reduced Glutathione (GSH) Content

GSH levels were assayed according to Teti and collaborators [56], with slight modifications. Samples (20% hematocrit), which were left untreated or exposed to 10 µM HgCl_2_ (4 h at 37 °C) or pre-incubated with 10 µM HT and/or HVA (15 min at 25 °C, with or without HgCl_2_ treatment), were centrifuged (Neya 16R, 4 °C, 1200× *g*, 5 min) and resuspended in isotonic solution. After treatments, the content of GSH was measured by Cayman’s GSH assay kit using an enzymatic recycling method with glutathione reductase. This assay is based on the oxidation of GSH by Ellman’s reagent DTNB, which produces oxidized glutathione (GSSG) and 3-thio-2-nitro-benzoic acid (TNB), absorbing at a wavelength of 412 nm. The amount of GSSG was calculated by the following formula: 1/2 GSSG = GSH total-GSH reduced. Results are expressed as a GSH/ GSSG ratio.

### 2.12. Experimental Data and Statistics 

All data are expressed as arithmetic mean ± standard error of the mean. For statistical analysis and graphics, GraphPad Prism (version 9.0, GraphPad Software, San Diego, CA, USA) and Excel (Version 2019, Microsoft, Redmond, WA, USA) software were used. Data normality was verified with the D’Agostino and Pearson Omnibus normality test. Significant differences between mean values were determined by one-way analysis of variance (ANOVA), followed by Bonferroni’s multiple comparison post-test or ANOVA with Dunnett’s post-test, as appropriate. Statistically significant differences were assumed at *p* < 0.05; (n) corresponds to the number of separate measurements.

## 3. Results

### 3.1. Evaluation of Red Blood Cell Shape

As depicted in Figure 2, incubation for 4 h at 37°C with 10 μM HgCl_2_ induced the morphological alteration of RBCs with 44.3% of acanthocytes (RBCs with surface blebs) detected by scanning electron microscopy analysis (SEM). However, in samples pre-treated for 15 min at 25 °C with 10 μM HT and/or HVA and then treated with 10 μM HgCl_2_, the percentage of morphologically altered cells was reduced to 3.5% and 2.9%, respectively (Table 1).

### 3.2. Oxidative Stress Assessment

#### 3.2.1. Evaluation of Intracellular ROS Levels

The evaluation of ROS species was carried out by flow cytometry in RBCs left untreated or, alternatively, exposed to 10 µM HgCl_2_ with or without pre-exposure to 10 µM HT and/or HVA for 15 min at 25 °C. Figure 3A shows the intracellular ROS levels at different time points (0, 30, 60, 120, 180, and 240 min after exposure to 10 µM HgCl_2_). 

Samples exposed to 10 µM HgCl_2_ showed a significant increase in ROS levels compared to the control samples. After 30 min, the levels of ROS increased by 50% in the 10 µM HgCl_2_-treated samples and remained unchanged over time. In Figure 4, the effect of olive oil polyphenols is also reported. In samples pre-exposed to 10 µM HT and/or HVA, 10 µM HgCl_2_ failed to significantly increase ROS levels, with these being unchanged compared to control values (Figure 3a). Of note, olive oil polyphenols alone did not significantly induce an increase in ROS levels.

#### 3.2.2. Measurement of Thiobarbituric Acid Reactive Substance (TBARS) Levels

The measurement of Thiobarbituric acid reactive substances (TBARSs) in RBCs is reported in Figure 3b. As expected, the TBARS levels of RBCs treated with 20 mM H_2_O_2_ (positive control) for 1 h were significantly higher with respect to those of RBCs left untreated (control). Moreover, the TBARS levels of RBCs treated with 10 µM HgCl_2_ for 4 h were significantly higher than those of RBCs left untreated (control). Importantly, in RBCs pre-treated with 10 µM HT and/or HVA and then exposed to 10 µM HgCl_2_, TBARS levels were significantly reduced compared to those measured in 10 µM HgCl_2_-treated RBCs. It is of note that olive oil polyphenols alone did not significantly affect TBARS levels.

#### 3.2.3. Total Sulfhydryl Group Content Measurement

Figure 3c shows the total content of sulfhydryl groups (µM TNB/µg protein) in RBCs left untreated or treated with either the oxidizing compound NEM (2 mM for 1 h, as the positive control), or 10 µM HgCl_2_ for 4 h with or without HT and/or HVA pre-treatment. As expected, exposure to NEM led to a significant reduction in the sulfhydryl group content. Sulfhydryl groups in 10 µM HgCl_2_-treated RBCs were also significantly reduced with respect to the control. Importantly, pre-treatment with 10 µM HT and/or HVA significantly restored sulfhydryl group total content in 10 µM HgCl_2_-treated RBCs (Figure 3b). Olive oil polyphenols alone did not significantly affect the total sulfhydryl group content (Figure 3b).

#### 3.2.4. Evaluation of Methemoglobin (MetHb) Levels

Figure 3d shows MetHb levels (%MetHb) measured in RBCs left untreated or treated with 10 µM HgCl_2_ (4 h at 37 °C) with or without pre-treatment with 10 µM HT and/or HVA (1 h at 25 °C), or, alternatively, treated with the well-known MetHb-forming agent NaNO_2_ (4 mM for 1 h at 25 °C). Methemoglobin levels measured after incubation with NaNO_2_ were significantly higher than those detected in RBCs left untreated (control). In parallel, MetHb levels measured following exposure to 10 µM HgCl_2_ were significantly higher than those measured in the control (left untreated). Pre-exposure to 10 µM HT and/or HVA significantly reduced the MetHb levels in 10 µM HgCl_2_-treated RBCs towards values that did not differ from control values. Olive oil polyphenols alone did not significantly affect the %MetHb.

### 3.3. Determination of Apoptotic Red Blood Cells

Figure 4 shows eryptosis levels measured in RBCs left untreated or treated with 10 µM HgCl_2_ (4 h at 37 °C) with or without pre-treatment with 10 µM HT and/or HVA (1 h at 25 °C). Regarding the percentage of RBCs in eryptosis, significant differences were detected after treatment with 10 µM HgCl_2_ for 4 h. However, pre-treatment with 10 µM HT and/or HVA significantly reduced eryptosis levels in 10 µM HgCl_2_-treated RBCs. Olive oil polyphenols alone did not significantly affect eryptosis.

### 3.4. Detection of Band 3 Protein Levels

Figure 5 shows B3p levels in RBCs incubated with 10 µM HgCl_2_ (4 h at 37 °C) with or without pre-treatment with 10 µM HT and/or HVA (1 h at 25 °C). Band 3 protein levels after treatments were not significantly different with respect to those determined in control RBCs.

### 3.5. Measurement of SO_4_^2−^ Uptake through Band 3 Protein 

Figure 6 reports the SO_4_^2−^ uptake as a function of time in RBCs left untreated (control) and in RBCs treated with 10 µM HgCl_2_ (4 h at 37 °C) with or without pre-treatment with 10 µM HT and/or HVA (1 h at 25°C). In control conditions, SO_4_^2−^ uptake progressively increased and reached equilibrium within 45 min (rate constant of SO_4_^2−^ uptake = 0.058 ± 0.001 min^−1^). Red blood cells treated with 10 µM HT and/or HVA alone showed a rate constant of SO_4_^2−^ uptake that was not significantly different with respect to the control. Conversely, the rate constant value (0.043 ± 0.001 min^−1^) in RBCs treated with 10 µM HgCl_2_ was significantly lower than in the control (*** *p* < 0.001). In RBCs pre-incubated with 10 µM HT (1 h at 25 °C) and then treated with 10 µM HgCl_2_ (1 h at 25 °C), the rate constant (0.058 ± 0.001 min^−1^) was significantly higher than that of RBCs treated with 10 µM HgCl_2_ (0.043 ± 0.001 min^−1^) while not being significantly different with respect to the control (Table 2). Similarly, in RBCs pre-incubated with 10 µM HVA (1 h at 25 °C) and then exposed to 10 µM HgCl_2_ (4 h at 37 °C), the rate constant (0.062 ± 0.001 min^−1^) was significantly different with respect to that of RBCs treated with 10 µM HgCl_2_ (0.043 ± 0.001 min^−1^) (Table 2). SO_4_^2−^ uptake was almost completely blocked by 10 µM DIDS applied at the beginning of incubation in SO_4_^2−^ medium (0.018 ± 0.001 min^−1^, *** *p* < 0.001, Table 2). Additionally, the SO_4_^2−^ amount internalized by HgCl_2_-treated RBCs after 45 min of incubation in SO_4_^2−^ medium was significantly lower than the control (Table 2), while in RBCs pre-incubated with 10 µM HT and/or HVA and then exposed to 10 µM HgCl_2_, it was not significant different with respect to control (Table 2). In DIDS-treated cells, the internalized SO_4_^2−^ amount (5.49 ± 2.50) was significantly lower than what was determined in both control and treated RBCs (*** *p* < 0.001, Table 2).

### 3.6. GSH/GSSG Ratio Measurement

Figure 7 shows the GSH/GSSG ratio measured in RBCs treated with 10 µM HgCl_2_ for 4 h at 37 °C with or without pre-treatment with 10 µM HT and/or HVA (15 min at 25 °C). The GSH/GSSG ratio measured after incubation with 10 µM HgCl_2_ was significantly lower than that detected in control RBCs. This effect can be associated with an increased GSSG abundance and/or decreased GSH concentration, both of which are indicative of an increase in cellular oxidative stress.

## 4. Discussion

In this study, the antioxidant capacity of the most important olive oil polyphenol metabolites (HT and/or HVA) on morphology and function on an HgCl_2_-induced oxidative stress model in human RBCs were investigated. With regard to the first aspect, the susceptibility of RBCs to 10 µM HgCl_2_ exposure was verified by scanning electron microscopy (SEM), revealing dramatic changes in the RBC shape. Such a finding is in line with what was already reported by Lim and collaborators [57] when they observed RBC shape changes from acanthocytes to echinocytes to spherocytes accompanied by micro-vesicle generation after prolonged exposure to low-dose HgCl_2_. In the present case, the typical biconcave shape was lost in a significant number of cells displaying surface blebs (acanthocytes, 10 µM HgCl_2_). Indeed, the percentage of acanthocytes was higher (44.3%) than that detected in untreated cells (3%). However, pre-treatment with 10 µM HT and/or HVA polyphenol extracts prevented the morphological changes (Figure 2c,d), with a reduction in the percentage of acanthocytes (Table 1). These data also corroborate Paiva-Martins and co-authors [22,58], who reported that treatment with both olive oil polyphenols protected RBC morphology from the oxidative damage induced by the radical initiator 2,2-azo-bis(2-amidinopropane) dihydrochloride or, alternatively, caused by H_2_O_2_ exposure. 

The study of RBC morphology is of great importance in the field of hemorheology [59,60]. Although RBC membranes are designed to withstand the mechanical stress of blood flow, these cells may respond to any form of damage by changing their morphology following changes in their structure or biochemical composition (the oxidation of sulfhydryl groups of membrane proteins, the oxidation of membrane fatty acid residues, or the oxidation of hemoglobin [61]). In this regard, the exposure to HgCl_2_ has also been associated with eryptosis [62,63]. By analogy with the apoptosis of nucleated cells, RBCs may undergo programmed cell death, which is characterized by cell shrinkage and cell membrane phospholipid scrambling. Eryptosis machinery includes the activation of redox-sensitive calcium-permeable cation channels, resulting in calcium entry [64]. The elevation of cytosolic calcium further activates RBC scramblase and calpain resulting in PS externalization and membrane blebbing, respectively. In this context, 10 µM HgCl_2_ exposure induced PS translocation to the external surface of the RBC membranes after 4 h of treatment (Figure 4). 

Specifically, HgCl_2_ treatment inhibited flippase, an enzyme that recovers PS into the inner leaflet of the cell membrane, and activated scramblase, an enzyme that alters lipid asymmetry in the cell plasma membrane [65], resulting in increased pro-coagulant activity and the removal of RBCs from the circulatory stream [57,66]. In this investigation, the antioxidant capacity of HT and/or HVA to prevent HgCl_2_-induced PS exposure and morphological changes in human RBCs was demonstrated (Figure 4). 

Since the oxidation of biological macromolecules, such as lipids and proteins, derives from the deleterious effects of ROS generated during normal cellular metabolism [67,68], intracellular ROS levels after exposure to 10 µM HgCl_2_ were initially evaluated. Our data showed that pre-treatment with 10 μM HT and/or HVA induced a decrease in ROS levels induced by treatment with 10 µM HgCl_2_ (Figure 3a). 

This finding is supported by many authors and suggests that dietary supplementation with regular polyphenol consumption has an antioxidant activity, with this being generally attributed to their ability to directly neutralize ROS, in particular in populations occupationally exposed to HgCl_2_ poisoning [22,29,69]. To better explain the mechanisms underlying the morphological changes, certain parameters related to oxidative stress assessment were monitored. Different phenomena, including the oxidation of membrane fatty acid residues, the oxidation of sulfhydryl groups of membrane proteins, or the oxidation of hemoglobin, could alter membrane properties and cell shape, thus causing a decrease in the deformability and a loss in membrane integrity. Oxidative modifications tend to stimulate the accumulation of oxidized lipids and proteins, which, when present in excess amounts, impair RBC architecture and function and result in the shortened life span of these cells. Due to oxidation stress, membrane polyunsaturated fatty acids of RBCs are damaged. This results in a steep increase in malondialdehyde (MDA) and thiobarbituric acid (TBA) levels, biomarkers currently used to reveal the oxidation of lipids under different experimental conditions [70]. The present data show that 15 min of pre-treatment with 10 µM HT and/or HVA prevented lipid peroxidation of membranes induced by HgCl_2_ treatment for 4 h (Figure 3b). However, ROS attack on lipids initiates a chain reaction, which leads to the generation of more ROS that can harm other cellular components, including proteins [71]. Protein oxidation is characterized by a simultaneous decrease in free amino and sulfhydryl groups. Thus, the measurement of the sulfhydryl group content of total proteins was also evaluated. Exposure to 10 μM HT and/or HVA protected RBC proteins from oxidative injury (Figure 3c). These data are in line with what has been previously demonstrated by other authors. For example, pre-incubation with HT and HVA inhibited H_2_O_2_-induced lipid-peroxidation in LLC-PK1, a porcine kidney epithelial cell line [72]. In addition, the effects of HT exposure on UVA-induced cell damages were investigated using a human melanoma cell line (M14) as a model system [73]. In UVA-irradiated M14 cells, HT extract prevented the uprise of typical markers of oxidative stress, such as TBARSs, protein oxidation, and an increase in ROS levels.

The biological effects of olive oil are related to its chemical matrix, which includes numerous phytochemicals components, such as flavonoids [74]. These molecules can directly neutralize ROS and/or attenuate lipid peroxidation by scavenging free radicals, thus sustaining the endogenous antioxidant system [26,75]. In a previous investigation [76], the potential protective role of the polyphenolic flavonoid compound quercetin (Q) on an oxidative stress model represented by RBCs treated with 20 mM H_2_O_2_ was evaluated. In this regard, TBARS levels, augmented following 20 mM H_2_O exposure (1 h), were completely restored by 10 µM Q pre-treatment (1 h). Similarly, the oxidation of protein sulfhydryl groups was totally restored after pre-treatment (1 h) with Q in RBCs treated with 20 mM H_2_O_2_ (1 h). Additionally, the protective effects of freeze-dried Açaì extract—the main chemical components present in the Açaí pulp are polyphenols—in a D-Galactose-induced aging model in human RBCs have been also reported [37]. 

Another major property of RBC oxidation is the clustering and/or the breakdown of B3p and the binding of oxidized hemoglobin (MetHb) to high affinity sites on B3p. Band 3 protein is an integral membrane protein that accounts for approximately 25% of the RBC membrane. It has several crucial functions and is composed by two rather different domains of a similar size [77]. In RBCs, HgCl_2_-induced oxidative damage resulted in the oxidation of the ferrous iron of hemoglobin to a ferric form given methemoglobin (MetHb), which is inactive as an oxygen transporter in the blood [7]. To better elucidate the molecular interaction between B3p and oxidized hemoglobin, MetHb levels were also evaluated. Our results indicated that exposure to 10 µM HgCl_2_ for 4 h increased the levels of MetHb in RBCs (Figure 3c). These modifications can start a cascade of structural and biochemical transformations, including the release of microparticles containing hemicromes and the clustering of B3p regions, as formerly demonstrated by other groups [23,24,69]. When the HgCl_2_-induced oxidation processes are advanced, these clusters can provide a recognition site for antibodies directed against aging cells, thus triggering the premature removal of senescent RBCs from the circulation at the end of their 120-day life span [78]. Interestingly, pre-treatment with 10 μM HT and/or HVA prevented HgCl_2_-induced MetHb production (Figure 3d), which was also observed when Q was applied after an oxidative stress increase (20 mM H_2_O_2_) [76]. 

One of the most interesting and still poorly investigated implications of HgCl_2_ toxicity is its impact on membrane transport systems. The Band 3 protein is mainly involved in the chloride–bicarbonate exchange. Firstly, the content of B3p was determined. No change in the protein content was detected under these experimental conditions (Figure 5). Secondly, to define the potential effect of HgCl_2_ on RBC functional activity, the SO_4_^2−^ uptake was measured by means of a validated method to assay anion exchange capability via B3p [79]. In RBCs incubated with 10 μM HgCl_2_, the rate constant for SO_4_^2−^ uptake was reduced compared to the control (Figure 6 Table 2), and, in parallel, the amount of internalized SO_4_^2−^ was significantly reduced. However, 15 min pre-treatment with 10 µM of HT and/or HVA completely restored the SO_4_^2−^ absorption rate constant as well as the amount of internalized SO_4_^2−^, thus confirming the beneficial effect of the olive oil polyphenols on B3p function. 

The evidence that B3p exhibits modifications in the rate constant for SO_4_^2−^ uptake following the exposure of RBCs to different oxidative stressors has been previously demonstrated. Specifically, H_2_O_2_ non-hemolytic (300 µM for 30 min) concentrations induced oxidative stress and provoked a decrease in the rate constant for SO_4_^2−^ uptake via B3p [43]. However, pre-treatment with melatonin ameliorated the reduction in the rate constant for SO_4_^2−^ uptake as well as the reduction in B3p expression levels observed following treatment with H_2_O_2_ [34]. Moreover, the reduction in B3p anion exchange efficiency caused by a mild oxidative stress (300 µM for 30 min) was prevented or attenuated by a short-term pre-incubation of RBCs with low H_2_O_2_ doses (10 µM for 30 min) [35]. This pre-treatment allows RBCs to adapt to a mild and transient oxidative stress and favors an increased tolerance to a successive stronger oxidant condition. Such an adaptative response, termed pre-conditioning and monitored by measuring B3p activity, did not involve B3p-related Tyr-phosphorylation pathways but was mediated by increased catalase activity. In summary, a reduction in the transport rate following H_2_O_2_ exposure is most likely linked to the formation of oxidizing hemoglobin after oxidative stress. Therefore, the different effect on the SO_4_^2−^ transport kinetics observed in a former study is not surprising. Indeed, in other models of oxidative stress obtained by exposing RBCs to high concentrations of D-Galactose and D-Glucose, the acceleration of the anion exchange through B3p, instead of a slowing, was detected [31,32]. It is tempting to speculate that such a two-sided effect on anion exchange velocity depends on the specific structure targeted by the stressors and on the possible underlying pathways. To name just a few examples, the acceleration of the transport rate in two former studies could be most likely linked to mechanisms other than oxidative stress, or, putatively, to the formation of glycated hemoglobin [31,80]. 

Finally, we directed our focus towards an assay of the endogenous antioxidant system. The underlying mechanism of HgCl_2_ toxicity is correlated with a high affinity of HgCl_2_ for sulfhydryl groups that can severely alter enzyme activities, including Glutathione [81]. The HgCl_2_ ions react with sulfhydryl groups of cysteine residue, thus depleting intracellular thiols [82]. A decrease in total sulfhydryl group content demonstrates redox imbalance and the perturbation of the thiol status of the cell (Figure 3b). Hence the GSH/GSSG ratio was evaluated. The obtained results confirmed that 10 µM HgCl_2_ treatment reduced the GSH/GSSG ratio. However, the pre-incubation of RBCs with 10 µM HT and/or HVA completely restored the redox balance (Figure 7). To support this evidence, in a rat model of Parkinson’s disease, HT extract also resulted in the preservation of the brain GSH/GSSG ratio after MPP^+^-induced oxidative injury [83]. Glutathione is an important intracellular non-enzymatic antioxidant that neutralizes many electrophiles and ROS [7]. Its depletion makes cells more prone to oxidative damage. To confirm this finding, a study performed by Lund and co-authors [84] suggests that HgCl_2_ increases the production of hydrogen peroxide and GSH depletion in renal mitochondria, thus causing oxidative damage in cell membranes, resulting in hemolysis.

## 5. Conclusions

In summary, exposure to HgCl_2_ induced increased oxidative stress in human RBCs. Indeed, the damaged RBCs may quickly be removed from circulation, thus leading to the shortened lifespan of RBCs, resulting in anemia. However, pre-treatment with HT and/or HVA avoided the formation of acanthocytes observed after exposure to HgCl_2_ prevented HgCl_2_-induced oxidative stress damage, including ROS production, lipid peroxidation, and sulfhydryl group oxidation of the total proteins on the plasma membrane. Moreover, HgCl_2_ exposure was associated with a reduction in the rate constant of SO_4_^2−^ uptake through B3p as well as MetHb production. Both alterations were attenuated by pre-treatment with HT and/or HVA. Importantly, pre-incubation with both polyphenols restored the GSH/GSSG ratio, resulting in the improvement of the endogenous antioxidant system.

In conclusion, the present investigation elucidates benefits deriving from the use of naturally occurring polyphenols against HgCl_2_-induced oxidative stress damage on a cellular level. Additionally, the results obtained here confirm that the measurement of the B3p anion exchange capability remains a suitable tool for monitoring the impact of oxidative stress on RBC homeostasis. Considering the involvement of oxidative stress damage in HgCl_2_-induced poisoning, new biomarkers with both diagnostic and monitoring potential are needed. Blood can be obtained from patients with minimally invasive procedures, with it reflecting the physiological status of peripheral tissues, and therefore it might represent a convenient source of biomarkers. In this light, future investigations are needed to clarify the signaling underlying the protective activity of HT and/or HVA on the interaction between B3p and cytoskeletal proteins such as ankyrin and spectrin and their potential post-translation modifications. These results are a step towards a better understanding of the biochemical mode of mercury-induced toxicity and cell damage. This will allow for appropriate methods to overcome the harmful effects of this extremely dangerous metal and widespread environmental pollutant to be devised.

## Figures and Tables

**Figure 1 cells-12-00424-f001:**
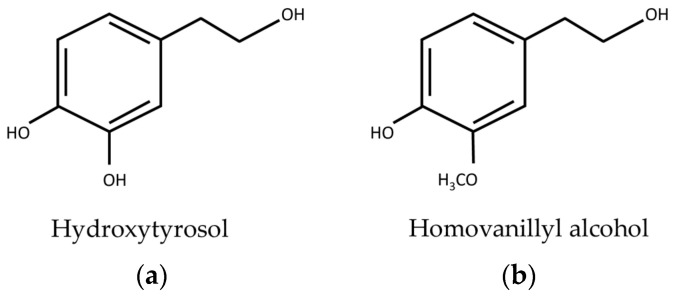
Chemical structures of (**a**) hydroxytyrosol (HT) and (**b**) homovanillyl alcohol (HVA).

**Figure 2 cells-12-00424-f002:**
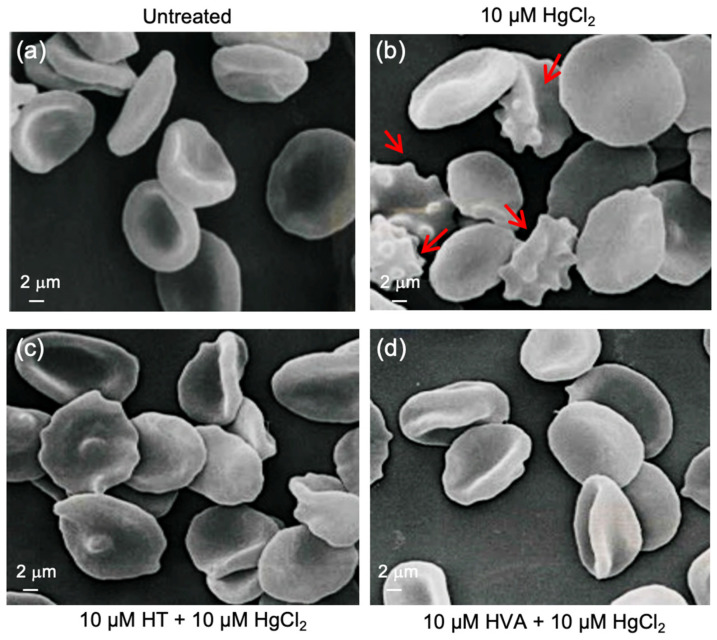
Red blood cell morphology evaluation. Representative scanning electron microscopy images showing RBCs with a typical biconcave form (**a**) left untreated or, alternatively, with surface blebs (**b**) (acanthocytes, red arrows) (10 μM HgCl_2_). (**c**) and (**d**) Pre-treatment with HT and/or HVA (10 µM) attenuated the morphological changes compared to HgCl_2_ treatment. Magnification 3500×.

**Figure 3 cells-12-00424-f003:**
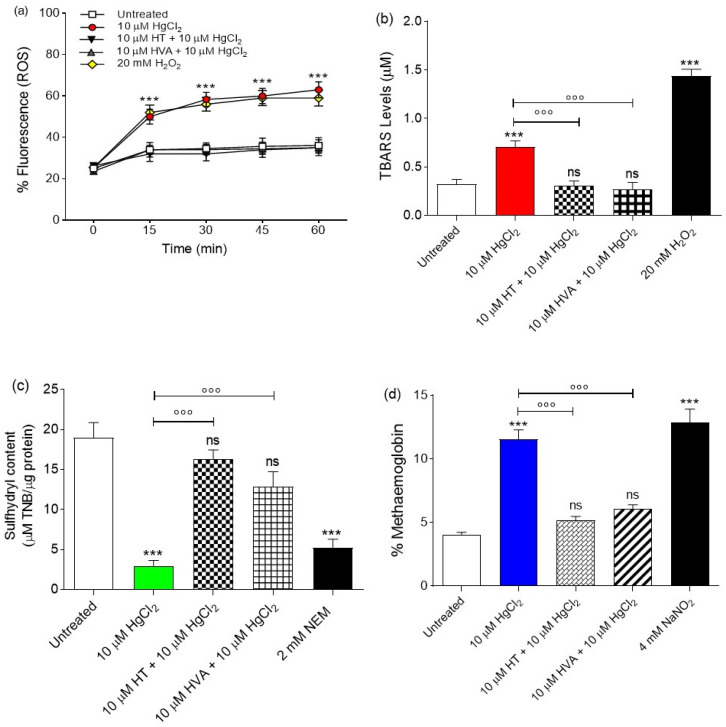
Determination of oxidative stress levels. (**a**) Detection of reactive oxygen species (ROS) levels by flow cytometry. Time course of ROS production in RBCs left untreated (control) or treated for 4 h with 10 µM HgCl_2_, with or without pre-exposure to 10 µM HT and/or HVA for 15 min. H_2_O_2_ was used as a positive control. ns, not statistically significant versus left untreated (control); *** *p* < 0.001 versus control, one-way ANOVA followed by Bonferroni’s post-hoc test (*n* = 8). (**b**) Detection of TBARS levels. TBARS levels (µM) in RBCs left untreated (control) or treated for 4 h with 10 µM HgCl_2_, with or without pre-exposure to 10 µM HT and/or HVA for 15 min. H_2_O_2_ was used as a positive control. ns, not statistically significant versus left untreated (control); *** *p* < 0.001 versus left untreated (control); °°° *p* < 0.001 versus 10 µM HgCl_2_, one-way ANOVA followed by Bonferroni’s post-hoc test (*n* = 11). (**c**) Sulfhydryl group content evaluation. Sulfhydryl group content (µM TNB/µg protein) in RBCs left untreated (control) and in RBCs treated for 4 h with HgCl_2_, with or without pre-exposure to 10 µM HT and/or HVA for 15 min. ns, not statistically significant versus left untreated (control); *** *p* < 0.001 versus left untreated (control); °°° *p* < 0.001 versus 10 µM HgCl_2_, one-way ANOVA followed by Bonferroni’s post-hoc test (*n* = 11). NEM was used as a positive control. ns, not statistically significant versus control; *** *p* < 0.001 versus control, one-way ANOVA followed by Bonferroni’s post-hoc test (*n* = 10). (**d**) Methemoglobin (% MetHb) content. Red blood cells were left untreated or incubated with 10 µM HgCl_2_, with or without pre-exposure to 10 µM HT and/or HVA for 15 min. NaNO_2_ (4 mM for 1 h) was used as positive control. ns, not statistically significant; *** *p* < 0.001 versus left untreated (control); °°° *p* < 0.001 versus 10 µM HgCl_2_, one way ANOVA followed by Bonferroni’s post-hoc test (*n* = 11).

**Figure 4 cells-12-00424-f004:**
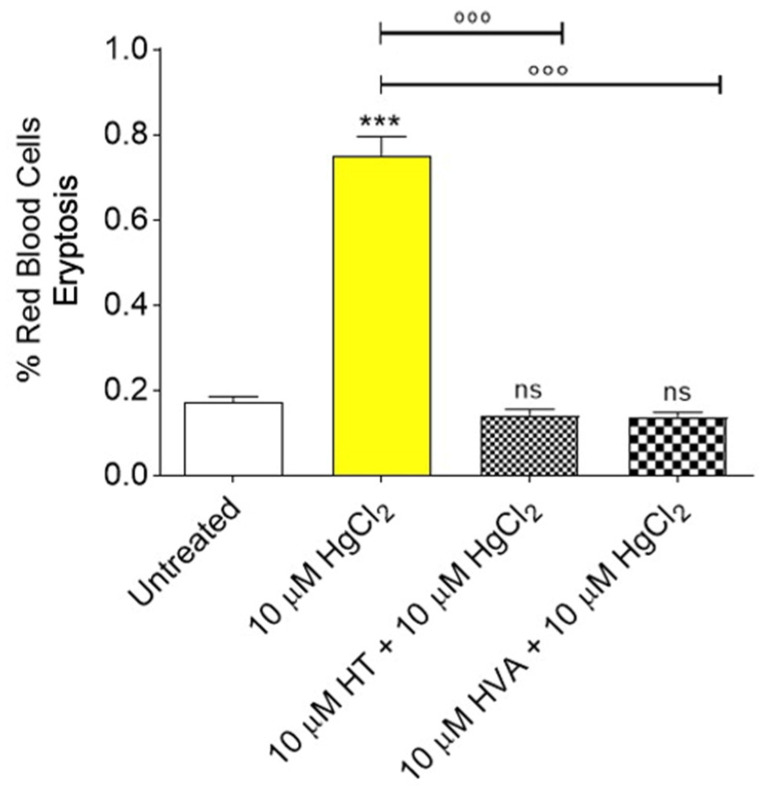
Detection of eryptosis by flow cytometry. Percentage of apoptotic RBCs positive to Annexin 5 and Trypan Blue detected in samples treated with 10 µM HgCl_2_ (4 h at 37 °C) with or without pre-treatment with 10 µM HT and/or HVA (1 h at 25 °C). ns, not statistically significant versus control; *** *p* < 0.001 versus left untreated (control); °°° *p* < 0.001 versus 10 µM HgCl_2_, one way ANOVA followed by Bonferroni’s post-hoc test (*n* = 11).

**Figure 5 cells-12-00424-f005:**
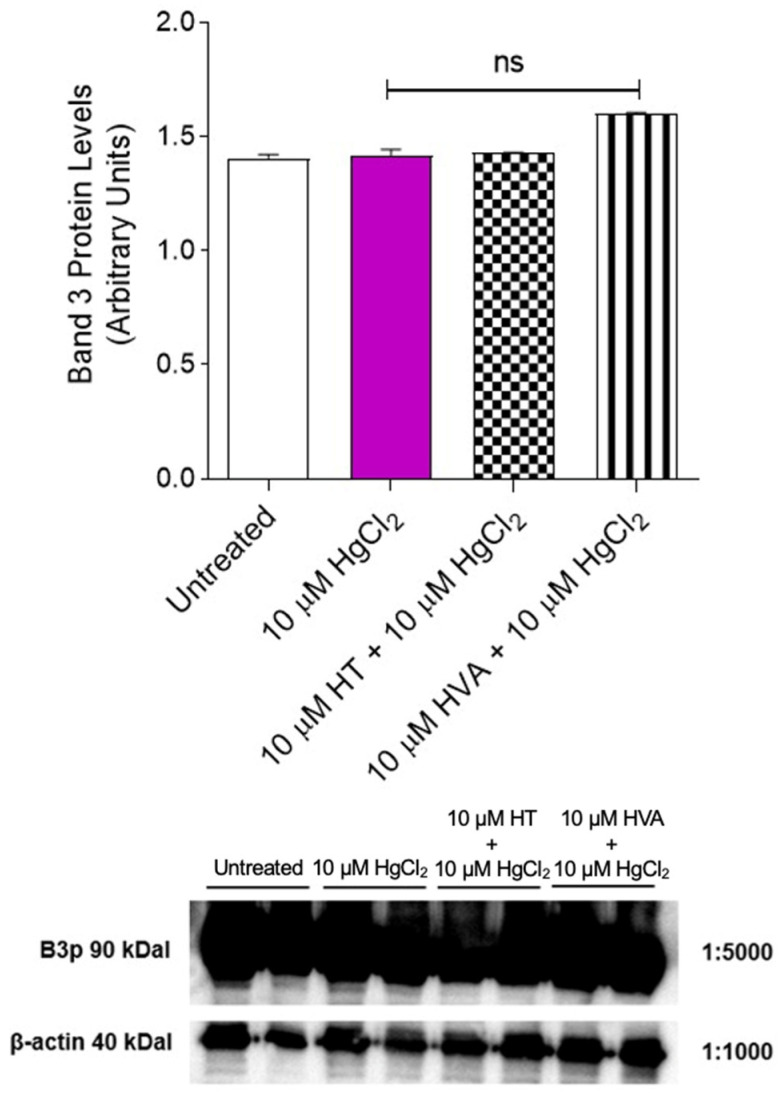
Band 3 protein expression levels measured in RBCs left untreated (control) or treated with 10 µM HgCl_2_ for 4 h at 37 °C with or without pre-exposure to 10 µM HT and/or HVA for 1 h at 25 °C, detected by Western Blotting analysis. ns, not significant versus left untreated (control), one-way ANOVA followed by Bonferroni’s multiple comparison post-hoc test (*n* = 3).

**Figure 6 cells-12-00424-f006:**
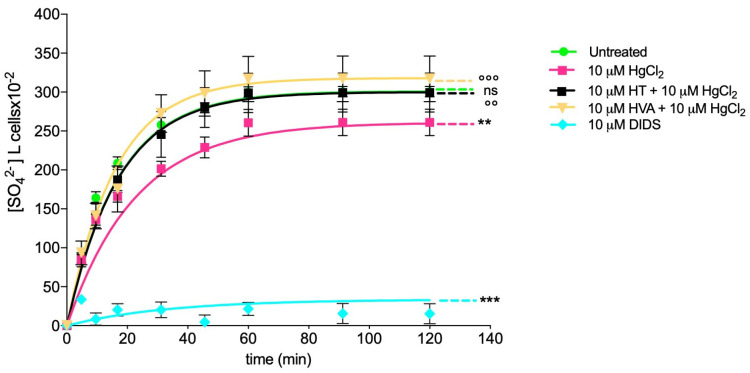
Time course of SO_4_^2−^ uptake. Red blood cells were left untreated (control) or treated with 10 µM HgCl_2_ for 4 h at 37 °C with or without pre-exposure to 10 µM HT and/or HVA for 1 h at 25 °C or exposed to 10 µM DIDS. ns, not statistically significant versus control; ** *p* < 0.01 and *** *p* < 0.001 versus control; °° *p* < 0.05 or °°° *p* < 0.001 versus 10 µM HgCl_2_, one way ANOVA followed by Bonferroni’s post-hoc test.

**Figure 7 cells-12-00424-f007:**
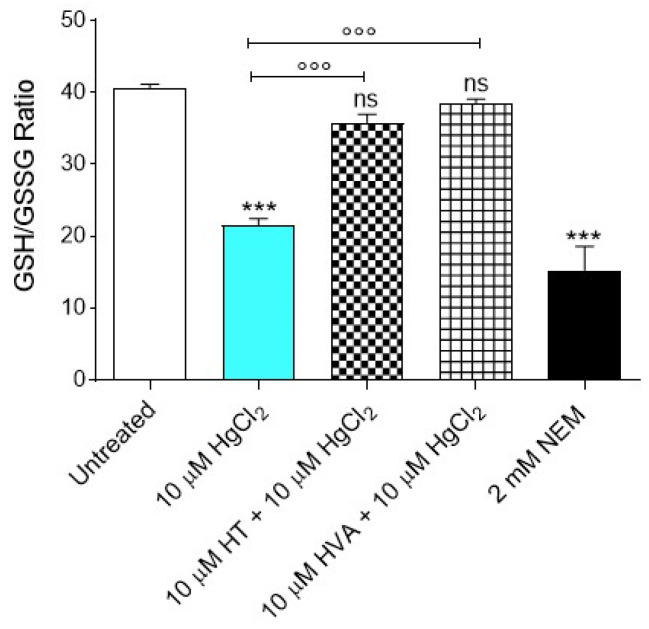
Estimation of the GSH/GSSG ratio measured in RBCs incubated for 4 h at 37 °C with 10 µM HgCl_2_ with or without pre-treatment with 10 µM HT and/or HVA (15 min at 25 °C). GSH, reduced glutathione; GSSG, oxidized glutathione. *** *p* < 0.001 versus left untreated (control) RBCs, °°° *p* < 0.001 versus 10 µM HgCl_2,_ one-way ANOVA followed by Bonferroni’s multiple comparison post-hoc test (*n* = 8).

**Table 1 cells-12-00424-t001:** Percentage of morphological alterations in RBCs left untreated (control) or treated as indicated. Data are presented as means ± S.E.M. from separate three independent experiments, where ns indicates not statistically significant versus left untreated (biconcave shape, acanthocytes); ^***^, *p* < 0.001 versus control (biconcave shape); ^°°°^, *p* < 0.001 versus control (acanthocytes), one-way ANOVA followed by Bonferroni’s multiple comparison post-hoc test.

Experimental Conditions	Biconcave Shape	Acanthocytes	n
**Untreated (control)**	94% ± 0.013	3% ± 0.011	5
**10 μM HgCl_2_**	55.7% ± 0.011 ^***^	44.3% ± 0.012 ^°°°^	5
**10 μM HT + 10 μM HgCl_2_**	96.5% ± 0.013 ^ns^	3.5% ± 0.009 ^ns^	5
**10 μM HVA + 10 μM HgCl_2_**	97.1% ± 0.007 ^ns^	2.9% ± 0.008 ^ns^	5

**Table 2 cells-12-00424-t002:** Rate constant of SO_4_^2−^ uptake and amount of SO_4_^2−^ trapped in RBCs left untreated (control) and RBCs treated as indicated. Results are presented as means ± S.E.M. from separate (*n*) experiments, where ns indicates not statistically significant versus left untreated or 10 µM HgCl_2_; ** *p* < 0.01; *** *p* < 0.001 versus control; °° *p* < 0.01 or °°° *p* < 0.001 versus 10 µM HgCl_2_, one-way ANOVA followed by Bonferroni’s multiple comparison post-hoc test.

Experimental Conditions	Rate Constant (min^−1^)	Time (min)	n	SO_4_^2−^ Amount Trapped after 45 min of Incubation in SO_4_^2−^ Medium (SO_4_^2−^) L Cells ×10^−2^
**Control**	0.058 ± 0.001	16.00	10	279 ± 17.43
**10 μM HgCl_2_**	55.7% ± 0.001 ^***^	22.23	10	228 ± 15.40 ^***^
**10 μM HT + 10 μM HgCl_2_**	96.5% ± 0.001 ^ns,°°^	16.95	10	280 ± 14.39 ^ns^
**10 μM HVA + 10 μM HgCl_2_**	97.1% ± 0.001 ^ns,°°°^	15.90	10	298 ± 17.70 ^**, ns^
**10 μM DIDS**	0.018 ± 0.001 ^***^	63.50	10	5.49 ± 3.60 ^***^

## Data Availability

The data that support the findings of this study are available from the corresponding author upon reasonable request.

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
