# Peer review of "Mercury Chloride Affects Band 3 Protein-Mediated Anionic Transport in Red Blood Cells: Role of Oxidative Stress and Protective Effect of Olive Oil Polyphenols"

_cells, 2023, doi:10.3390/cells12030424_

Round 1
Reviewer 1 Report
Dear Editor, dear authors
I have read the manuscript entitled “Mercury Chloride Affects Band 3 Protein-Mediated Anionic Transport in Red Blood Cells: Role of Oxidative Stress and Protective Effect of Olive Oil Polyphenols” written by Pasquale Perrone, Sara Spinelli, Gianluca Mantegna, Rosaria Notariale, Elisabetta Straface, Daniele Caruso, Giuseppe Falliti, Angela Marino, Caterina Manna, Alessia Remigante* and Rossana Morabito.
In the manuscript the authors report an investigation on the effect of natural antioxidants (hydroxytyrosol and homovanillyil alcohol – HT and HVA) on the oxidative pattern induced in RBCs by administration of HgCl2 and found that 10 minutes pre-treatments with selected polyphenols restored the antioxidant systems and protect against oxidative stress and development of morphological alterations.
The investigation have been conducted correlating Scanning Electron Microscopy imaging to the data from several biochemical/biophysical assays.
The paper is well written, the topic is interesting and contains several intriguing information both under the purely scientific and (potentially) applicative point of view. Yet, I believe there is still some work to do before publication.
Below are my suggestions for the entire text
Introduction:
The introduction is well written and contains all the necessary information to properly read and understand the paper
Materials and method:
What was the final concentration of DMSO employed for delivering the HT, HVA and DIDS (i.e. the actual DMSO concentration in the samples) and how the authors get rid of this chemical before the mercury exposure?
I have a concern on the determination of MetHb. As far as I understand, the authors lysed 25 uL of RBCs in an hypotonic buffer followed by ultracentrifugation, but they do not preliminary get rid of already lysed cell present in the sample. In practice this could cause the consideration of a part of extracellular Hb (obviously more oxidized) together with a majority of intracellular Hb (which is the part they want to evaluate). It seems to me that an evaluation (and, if adequate, a post processing correction) could require at least a measure of the intrinsic lysis in the samples after the chosen treatment and before the lysis. That could result in some % of unwanted extracellular Hb unwillingly mixed with the intracellular arising from the cell lysis and that, in my opinion, should be ruled out. Alternatively, a centrifugation and resuspension BEFORE the lysis of the 25 uL could be considered.
Results:
Figure 2 and table 1: Effect on the achantocytes abundance is interesting and deserve to be discussed, in the context of the paper’s result, in term of what kind of biochemical alteration can produce such a specific morphology. Please comment in the text.
Is this the only effect the author observed on the erythrocytes’ morphologies?
I cannot evaluate Figure 3 (at least in my copy). Panel d, describing metHb, is out of the page and, in general the four subpanels are a bit too small.
I suggest to place panels a,b in an upper row and panels c,d in a lower. This way they should result larger and more readable (this is mandatory).
Please revise the English between the lines 398-418
Figure 6 is very interesting.
Table 2: the data reported for the DIDS inhibited cells, although formally correct, refers to a data point objectively very small even in the context of the data trend.
How many times the authors’ repeated the experiment (please, add in the text)? Do they always found a decrease after 45 minutes in DIDS treated cells?
Lines 372-379
The question on the appropriateness on the term apoptosis for RBCs has long been debated and criticized in the literature. I suggest (is just a suggestion) the authors to consider the use of the more comfortable term eryptosis, although they discussed on this ambiguity in the discussion section.
Conclusion:
The section is an appropriate and synthetic summary of the most relevant results.
Although the paper is well written, designed in a clear way and accurately conducted, my concern is that the part of the document related to figure 3 cannot be thoroughly evaluated.
Thus, before publication, I recommend to revise the figure and to consider the recommendation and concerns on the whole text, that I reported above.
Author Response
- Point by point reply to the Reviewer 1
Dear Editor,
Dear Authors,
I have read the manuscript entitled -Mercury Chloride Affects Band 3 Protein-Mediated Anionic Transport in Red Blood Cells: Role of Oxidative Stress and Protective Effect of Olive Oil Polyphenols- written by Pasquale Perrone, Sara Spinelli, Gianluca Mantegna, Rosaria Notariale, Elisabetta Straface, Daniele Caruso, Giuseppe Falliti, Angela Marino, Caterina Manna, Alessia Remigante* and Rossana Morabito.
In the manuscript the authors report an investigation on the effect of natural antioxidants (hydroxytyrosol and homovanillyil alcohol –HT and HVA) on the oxidative pattern induced in RBCs by administration of HgCl2 and found that 15 minutes pre-treatments with selected polyphenols restored the antioxidant systems and protect against oxidative stress and development of morphological alterations. The investigation have been conducted correlating Scanning Electron Microscopy imaging to the data from several biochemical/biophysical assays. The paper is well written, the topic is interesting and contains several intriguing information both under the purely scientific and (potentially) applicative point of view. Yet, I believe there is still some work to do before publication.
We thank the Reviewer for the overall positive evaluation.
Below are my suggestions for the entire text.
Introduction:
The introduction is well written and contains all the necessary information to properly read and understand the paper.
We thank the Reviewer for this positive evaluation.
Materials and method:
What was the final concentration of DMSO employed for delivering the HT, HVA and DIDS (i.e. the actual DMSO concentration in the samples) and how the authors get rid of this chemical before the mercury exposure?
We thank the reviewer for this suggestion. Both ethanol and DMSO never exceeded 0.001% v/v in the experimental solutions and were previously tested on RBCs to exclude haemolytic damage (data not shown).
I have a concern on the determination of MetHb. As far as I understand, the authors lysed 25 uL of RBCs in an hypotonic buffer followed by ultracentrifugation, but they do not preliminary get rid of already lysed cell present in the sample. In practice this could cause the consideration of a part of extracellular Hb (obviously more oxidized) together with a majority of intracellular Hb (which is the part they want to evaluate). It seems to me that an evaluation (and, if adequate, a post processing correction) could require at least a measure of the intrinsic lysis in the samples after the chosen treatment and before the lysis. That could result in some % of unwanted extracellular Hb unwillingly mixed with the intracellular arising from the cell lysis and that, in my opinion, should be ruled out. Alternatively, a centrifugation and resuspension BEFORE the lysis of the 25 uL could be considered.
We thank the reviewer for this suggestion. During experiments, we performed a centrifugation and resuspension before lysing RBCs. In the full text, a sentence has been added.
Results:
Figure 2 and table 1: Effect on the acanthocytes abundance is interesting and deserve to be discussed, in the context of the paper’s result, in term of what kind of biochemical alteration can produce such a specific morphology. Please, comment in the text. Is this the only effect the author observed on the erythrocytes’ morphologies?
We thank the reviewer for this suggestion. A sentence has been added in the full text. The morphology of the circulating erythrocytes plays an important role on the rheological properties of the blood, and changes in morphology can lead to decreased deformability and/or increased aggregation. These cells may respond to any form of stressors by changing their morphology following changes in their membrane and/or biochemical composition. Certain phenomena such as the oxidation of sulfhydryl groups of membrane proteins, oxidation of membrane fatty acid residues, or oxidation of hemoglobin could alter membrane properties and the cell shape. Moreover, oxidation of biological macromolecules is also induced by the deleterious effects of ROS generated during cellular metabolism.
I cannot evaluate Figure 3 (at least in my copy). Panel d, describing metHb, is out of the page and, in general the four subpanels are a bit too small. I suggest to place panels a, b in an upper row and panels c, d in a lower. This way they should result larger and more readable (this is mandatory).
We thank the reviewer for this suggestion. Done.
Please revise the English between the lines 398-418.
We thank the reviewer for this suggestion. Done.
Figure 6 is very interesting.
We thank the Reviewer for this positive evaluation.
Table 2: the data reported for the DIDS inhibited cells, although formally correct, refers to a data point objectively very small even in the context of the data trend. How many times the authors repeated the experiment (please, add in the text)?
We thank the reviewer for this remark. DIDS is an irreversible anion exchange inhibitor that covalently bounds K851 and K539 residues of TM8, between core and gate domains (Reithmeier et al., 2016). As by our demonstrated by our previous investigations, DIDS inhibits SO42⁻ uptake through B3p. Moreover, experimental data (n=10) are validate by statistical analysis.
Do they always found a decrease after 45 minutes in DIDS treated cells?
We thank the reviewer for this query. Generally, we observed this trend after 45 minutes in DIDS treated cells. However, no statistical significant differences have been reported among analysed time points (10, 15, 30, 45, 60, 90 and 120 min).
Lines 372-379.
The question on the appropriateness on the term apoptosis for RBCs has long been debated and criticized in the literature. I suggest (is just a suggestion) the authors to consider the use of the more comfortable term eryptosis, although they discussed on this ambiguity in the discussion section.
We thank the reviewer for this suggestion. Done.
Conclusion: the section is an appropriate and synthetic summary of the most relevant results.
We thank the Reviewer for this positive evaluation.
Although the paper is well written, designed in a clear way and accurately conducted, my concern is that the part of the document related to figure 3 cannot be thoroughly evaluated. Thus, before publication, I recommend to revise the figure and to consider the recommendation and concerns on the whole text, that I reported above.
We thank the Reviewer for this positive evaluation. We added the Figure 3d in the full text.

Reviewer 2 Report
In this manuscript, entitled -Mercury Chloride Affects Band 3 Protein-Mediated Anionic Transport in Red Blood Cells: Role of Oxidative Stress and Protective Effect of Olive Oil Polyphenols- the purpose was to examine the protective capacity of HT and its metabolite HVA in a model of oxidative stress represented by human RBCs treated with a non-haemolytic concentration of HgCl2. The scientific work is very interesting, however, some minor problems, as indicated below, should be addressed before the document can be considered for publication. Here, I present all my comments in detail, but my global consideration is almost positive.
Minor revision:
-How does mercury address into red blood cells? The authors should add more information in the text.
-The accumulation of ROS under oxidative stress conditions results in the induction of lipid peroxidation and glycoxidation reactions, which leads to the elevated endogenous production of reactive aldehydes and their derivatives such as glyoxal, methylglyoxal (MG), malonic dialdehyde (MDA), and 4-hydroxy-2-nonenal (HNE) giving rise to advanced lipid-oxidation and glycation end products (ALEs and AGEs, respectively). Both ALEs and AGEs play key roles in cellular response to oxidative stress stimuli through the regulation of a variety of cell signaling pathways. Then, although this scientific work only summarizes the effects of Olive Oil Polyphenols on oxidative stress increase in the human RBCs treated with mercury, I suggest to introduce in the "Introduction" some knowledges regarding ALEs. Did authors investigate this aspect? They think they will be able to study it.
-The authors should modify Figure 4 (significant differences).
-Expand DIDS in the methods and provide IC50 of this agent.
-The exact role of SO42- uptake via Band 3 protein in human RBCs is unclear and not well understood. The mechanisms previously reported should be explained and how measuring SO42- provides an idea about the Band3 activity. Is the activity reported in RBCs of other species?
- 482-483 Since oxidation of biological macromolecules, such as lipids and proteins, derives from the deleterious effects of ROS generated during normal cellular metabolism or drugs, you can cite some articles with drugs affecting the oxidation of erythrocytes.
Author Response
In this manuscript, entitled -Mercury Chloride Affects Band 3 Protein-Mediated Anionic Transport in Red Blood Cells: Role of Oxidative Stress and Protective Effect of Olive Oil Polyphenols- the purpose was to examine the protective capacity of HT and its metabolite HVA in a model of oxidative stress represented by human RBCs treated with a non-haemolytic concentration of HgCl2. The scientific work is very interesting, however, some minor problems, as indicated below, should be addressed before the document can be considered for publication. Here, I present all my comments in detail, but my global consideration is almost positive.
We thank the Reviewer for the overall positive evaluation.
Minor revision:
-How does mercury address into red blood cells? The authors should add more information in the text.
We thank the reviewer for this suggestion. A reference has been added in the text.
-The accumulation of ROS under oxidative stress conditions results in the induction of lipid peroxidation and glycoxidation reactions, which leads to the elevated endogenous production of reactive aldehydes and their derivatives such as glyoxal, methylglyoxal (MG), malonic dialdehyde (MDA), and 4-hydroxy-2-nonenal (HNE) giving rise to advanced lipid-oxidation and glycation end products (ALEs and AGEs, respectively). Both ALEs and AGEs play key roles in cellular response to oxidative stress stimuli through the regulation of a variety of cell signaling pathways. Then, although this scientific work only summarizes the effects of Olive Oil Polyphenols on oxidative stress increase in the human RBCs treated with mercury, I suggest to introduce in the "Introduction" some knowledges regarding ALEs. Did authors investigate this aspect? They think they will be able to study it.
We thank the reviewer for this suggestion. In the next experimental design, we will investigate also this aspect.
-The authors should modify Figure 4 (significant differences).
We thank the reviewer for this suggestion. Done.
-Expand DIDS in the methods and provide IC50 of this agent.
We thank the reviewer for this suggestion. A reference has been added in the text.
-The exact role of SO42- uptake via Band 3 protein in human RBCs is unclear and not well understood. The mechanisms previously reported should be explained and how measuring SO42- provides an idea about the Band3 activity. Is the activity reported in RBCs of other species?
We want to thank the Reviewer for raising this point. In physiological conditions, Band 3 protein (B3p) favors chloride/bicarbonate (Cl-/HCO3-) electroneutral exchange across the plasma membrane. Since one of the peculiar features of erythrocytes is to rapidly exchange anions, it has been postulated that SO42- exchange may share a common mechanism with Cl- and HCO3- (Cabantchik et al., 1978), with the advantage that a turbidimetric method may more easily reveal the presence of transported SO42- than Cl-. In this regard, the rate constant for SO42- transport can be determined by a turbidimetric method, aimed at quantifying SO42- internalized through B3p as a function of time. Such in vitro technique, though not resembling the in vivo environment of erythrocytes, is a simple way to assess B3p anion exchange capability, as previously demonstrated by Jennings, 1976, Romano & Passow, 1984, Markovich, 2001, Romano et al., 2002, Teti et al., 2005 and Gugliotta et al., 2012.
-482-483. Since oxidation of biological macromolecules, such as lipids and proteins, derives from the deleterious effects of ROS generated during normal cellular metabolism or drugs, you can cite some articles with drugs affecting the oxidation of erythrocytes.
We thank the reviewer for this suggestion. Two references have been added in the text.

Reviewer 3 Report
The authors aims at evaluating the protective capacity of HT and its metabolite HVA in a model of oxidative stress represented by human RBCs treated with a non-haemolytic concentration of HgCl2. The manuscript is written, organized, very informative and timely, supported by tables and figures.
Minor Revision
- Line 215. The authors should add the protein concentration (µg/µL) in Materials and Methods. Bradford Assay is used to measure the concentration of total protein in these samples? The authors should add this protocol in Materials and Methods.
- Did authors perform experiments to assay the effects (alone) of HT and/or HVA without the HgCl2 incubation?
- Figure 3d is missing. The authors should add it in the full text.
Author Response
- Point by point reply to the Reviewer 3
The authors aims at evaluating the protective capacity of HT and its metabolite HVA in a model of oxidative stress represented by human RBCs treated with a non-haemolytic concentration of HgCl2. The manuscript is written, organized, very informative and timely, supported by tables and figures.
We thank the Reviewer for the overall positive evaluation.
Minor Revision
Line 215. The authors should add the protein concentration (µg/µL) in Materials and Methods. Bradford Assay is used to measure the concentration of total protein in these samples? The authors should add this protocol in Materials and Methods.
We thank the reviewer for this suggestion. Done.
Did authors perform experiments to assay the effects (alone) of HT and/or HVA without the HgCl2 incubation?
We thank the reviewer for this suggestion. RBCs have been treated with HT and/or HVA (alone) without the HgCl2 incubation, in order to investigate their -harmful- potential effects. No relevant effects have been reported. Thus, these data have not been shown in the full text.
Figure 3d is missing. The authors should add it in the full text.
We thank the reviewer for this suggestion. Done.

Round 2
Reviewer 1 Report
Dear Editor, dear Authors
I've read the revised version of the manuscript and the authors' response to my previous comments.
The authors satisfactorily answered to all my comments and amended the text accordingly.
I have no further comment and, as far as I'm concerned, the paper can be published as is.